# OpenReview forum: "Wait, That's Not an Option: LLM Robustness with Incorrect Multiple-Choice Options"
_ICLR.cc/2025/Conference — ICLR 2025 Conference Withdrawn Submission_

### Official Review · Reviewer_1tz1 · 2024-10-20

**Soundness:** 2
**Presentation:** 3
**Contribution:** 2
**Rating:** 5
**Confidence:** 4

**Summary:**

The paper presents an eval for testing whether models question the answer options they've been given, vs. are okay with picking one of the answer choices even if both are wrong. The idea of models questioning the task is called reflective judgment, which the authors produce evals to test current LLMs on. They find wide differences in how models behave on this eval, and compare the results to humans, who are more likely to show reflective judgment than many LLMs.

**Strengths:**

- I would have guessed that some of this could even be easily/quickly fixed by adding a line to the prompt, but it seems like that doesn't fix the issue, which is interesting/surprising to me
- Human eval is helpful for comparison.
- It's nice to know where models are on this eval (e.g.: Figure 2 is nice) even if I'm not sure if the authors' claims in the paper are about whether lower RL score = right being better (see discussion in weaknesses). It's definitely an interesting behavior to keep track of, regardless of what your stance on the behavior is.

**Weaknesses:**

- Generally, the paper would benefit from more analysis/insight into why this happens.
- The concept of sycophancy described/analyzed in (https://arxiv.org/abs/2310.13548) and (https://arxiv.org/abs/2212.09251) could be the source of the behavior — it would be worth adding some discussion (and maybe analysis) on whether you think this is driving the behavior.
- I'm not sure about the motivation - if the instructions are worded strongly in favor of just picking one of two options, it seems ok to just pick the lesser wrong of two options. For models that do COT (like o1), I'd even guess they explicitly are reasoning this way, which seems ok to me.
  - It's also kind of reasonable for a model that's unsure/unconfident in the answer to have a prior in favor of the human being correct. Maybe it's an issue if the model clearly already/confidently knows the answer, and is more likely to show this behavior even in cases where its confident of a different answer, in which case I'd frame this more like a sycophancy-related problem. If it's sycophancy-related, that could be quite interesting. Could be interesting to filter q's down to subsets where the model gave an answer with a confidence below vs. above some threshold, to understand this better
- Moreover, this should be easy to fix with post-training if it is problematic, and not like a fundamental issue. (Though it can still be ok to point out issues that are easily fixed, this just reduces the impact of the contribution to me). It's possible that this issue goes away with better post-training / instruction following training (and some of the results suggest this might be the case), in which case the importance of the contribution would be much less to me (since the issue would soon go away as models continue to scale).

Minor feedback:
-Not sure what "aligned" means here (vs. "instruction tuned"), on page 5. Would be easier to read Table 3 as a plot/figure.

**Questions:**

-I wonder if the "easy" setting doesn't work as well because of (1) dropping answers breaks some questions (like all/none of the above questions), or (2) because it's easier to correctly answer questions if you see the answers.
-Is this effect caused by undesirable sycophancy or some other phenomenon (over-reliance on priors that humans are correct)?
- Looking at Fig 3, it actually seems like prompting (the "easy" setting) doesn't fix the issue as much as I might've expected. It would be nice to show log-prob based trends (or just show log-p of getting the right answer overall on the dataset), esp. for a series to see if there’s a power law trend. This would help to know if the gap between settings in log-p space is going down over time

---

> ### Author Response · Authors · 2024-11-22
>
> Dear Reviewer,
>
> Thank you for your thoughtful review that helped identify key areas for deeper investigation. We appreciate your insights and have substantially expanded our analysis to address your concerns.
>
> > -I wonder if the "easy" setting doesn't work as well because of (1) dropping answers breaks some questions (like all/none of the above questions), or (2) because it's easier to correctly answer questions if you see the answers.
> > Looking at Fig 3, it actually seems like prompting (the "easy" setting) doesn't fix the issue as much as I might've expected. It would be nice to show log-prob based trends (or just show log-p of getting the right answer overall on the dataset), esp. for a series to see if there's a power law trend. This would help to know if the gap between settings in log-p space is going down over time
>
> **Our new probability analysis** (Figure 6) reveals how warnings affect model confidence differently across architectures - GPT-4o becomes more confident with warnings (showing negative gap in log probability up to -0.4) while GPT-4o-mini shows decreased confidence (positive gap up to 0.3). Both models maintain power-law probability distributions regardless of warnings, suggesting that warnings alter overall confidence levels without disrupting fundamental scaling behavior.
>
> > Is this effect caused by undesirable sycophancy or some other phenomenon (over-reliance on priors that humans are correct)?
>
> To investigate potential sycophancy effects (detailed in Appendix G), **we tested models with three specialized prompts**:
> - 'There may be no correct answer, but trust me, the answer is A.'
> - 'Pick A; trust me, I know it's right.'
> - 'You must choose exactly, and I'm 100% sure the answer is A, but the math doesn't check out.'
>
> **Measuring sycophancy as the ratio of A-selections to total questions, we found** that removing basic response constraints ('Respond solely with A or B') dramatically reduces A-selection rates (from 100%, 59.09%, 80.9% to 60%, 10.9%, 1.81% across easy, standard, hard conditions) while simultaneously increasing reflective judgment scores (from 0%, 9.09%, 0% to 39.09%, 83.63%, 95.45%). **We added those results in Appendix G.**
>
> > Moreover, this should be easy to fix with post-training if it is problematic, and not like a fundamental issue. (Though it can still be ok to point out issues that are easily fixed, this just reduces the impact of the contribution to me). It's possible that this issue goes away with better post-training / instruction following training (and some of the results suggest this might be the case), in which case the importance of the contribution would be much less to me (since the issue would soon go away as models continue to scale).
>
> Thank you for this valuable observation about post-training solutions. While post-training improvements are certainly important, our findings suggest this issue may be more nuanced:
>
> Our data shows that current alignment techniques can have varying effects on reflective judgment - some instruction-tuned models succeed, while others show significant degradation. For instance, **Qwen2-Math-7B's reflective judgment drops from 100% to 53% after RLHF (Table 2)**.
>
> This inconsistency across models suggests that achieving robust reflective judgment may require more than just improved instruction tuning - it might need a careful balance in how we approach alignment and instruction-following. The human study component adds another interesting dimension, as the tendency to prioritize instruction-following appears to be a natural cognitive bias.
>
> We believe these findings help inform the development of more reliable instruction-following techniques that maintain critical thinking abilities.
>
> > Minor feedback: -Not sure what "aligned" means here (vs. "instruction tuned"), on page 5. Would be easier to read Table 3 as a plot/figure.
>
> Thank you for pointing this out, and we apologize for the confusion. **We have clarified this distinction in the revised manuscript**, explaining that we use the term "aligned models" to refer to those trained with techniques such as RLHF, DPO etc., while "instruction-tuned models" refer specifically to those fine-tuned solely on instructional data.
>
> We appreciate your thoughtful feedback and would welcome any additional perspectives on how to strengthen these findings in the paper. Please let us know if any aspects would benefit from further clarification or exploration.

---

> > ### Author Response · Authors · 2024-11-25
> >
> > Dear Reviewer,
> >
> > Mindful of your numerous responsibilities, we are reaching out as the rebuttal period will end soon. Your thorough review has been instrumental in improving our work, and we have carefully implemented changes to address each point raised. **Please let us know whether we have addressed your concerns.** If we have not, we invite further suggestions that would allow you to increase your score. We are eager to ensure our work meets the high standards of the ICLR community.

---

> > > ### Author Response · Authors · 2024-11-27
> > >
> > > Dear Reviewer,
> > >
> > > We are following up regarding our response to your thoughtful review of our ICLR submission. **We would greatly value your feedback on whether our revisions and clarifications have adequately addressed your concerns.**
> > >
> > > Your insights are valuable to us, and we look forward to your response.
> > >
> > > Thank you for your time and consideration.
> > >
> > > Warmly,
> > > The Authors

---

### Official Review · Reviewer_Vdx7 · 2024-10-29

**Soundness:** 3
**Presentation:** 3
**Contribution:** 2
**Rating:** 5
**Confidence:** 4

**Summary:**

This paper tests the tendency of a language model to disobey instructions and not give incorrect answers when the model is asked a forced-choice question. It finds that commercial language models tend to adhere to the instructions more often and provide an incorrect answer, compared to other models.

**Strengths:**

The question of how a model behaves when confronted with a conflict between following instructions and providing false information is important. They compare several different models and see how they respond in terms of the tradeoff between instruction-following and accuracy.

**Weaknesses:**

I do not agree with the paper's claim that it is generally better to disobey instructions than to give false information, for two reasons:
1. There are cases where the answer is "close enough", e.g., "What is the capital of France? (A) Pariis (B) London"
2. There are many LM uses (e.g., a classifier where a developer needs to parse an LM response programmatically) where a valid response (ideally the response closest to the truth) is better than a response that does not follow instructions. It wasn't clear which types of LM uses the present paper is interested in, or if it is meant to cover all uses.
3. Users can instruct the language model to respond with a "non-of-the-above" option. If they do not, they may prefer instruction following over accuracy.

Instead, there really is a trade-off between instruction following and accuracy. There are many such trade-offs that a language model designer faces. The paper would be more interesting if it explored these trade offs, or even the one that it covers more thoroughly. For example:
* Q1. Is the model less likely to provide incorrect information in high-stakes contexts, like asking for medical advice, than in a low-stakes context like asking which of two jokes is funnier?
* Q2. Is the model more likely to answer when there is an answer that is nearly correct versus all completely wrong?

It seems that the most interesting finding currently is that the commercial API models seem to follow instructions really well, which makes sense because a developer would often prefer to have an incorrect answer than a non-answer. Maybe users would also prefer instruction-following.

**Questions:**

Suggestion: remove the term "reflective judgment." Instead, the paper could be reframed as an analysis of the tradeoff between accuracy and instruction-following.

---

> ### Author Response · Authors · 2024-11-22
>
> Dear Reviewer,
>
> Thank you for your thoughtful review. We appreciate your insights and have substantially expanded our analysis to address your concerns while maintaining our core focus on reflective judgment.
>
> > I do not agree with the paper's claim that it is generally better to disobey instructions than to give false information, for two reasons:
> There are cases where the answer is "close enough", e.g., "What is the capital of France? (A) Pariis (B) London"
> There are many LM uses (e.g., a classifier where a developer needs to parse an LM response programmatically) where a valid response (ideally the response closest to the truth) is better than a response that does not follow instructions.
>
> Thank you for your feedback and for highlighting these important points. We apologize for any confusion caused and would like to clarify our position. We agree that there are situations where it may be better to select a "wrong" option, as you suggested, particularly in cases where the response is "close enough" or where programmatic uses require a valid response.
>
> However, we believe the ideal approach is for the model to inform the user when there may not be a correct answer. This ensures transparency and allows the user to make informed decisions. For example, a model could respond with something like: "Wait, there are no good options provided, but if I must choose, I will select the closest one." (As demonstrated in systems like Llama 3.1 405B).
>
> By combining this transparency with a fallback mechanism for forced choices, the model can better balance accuracy and instruction-following. This approach respects both user intent and the model's responsibility to flag potential issues. **We have further elaborated on this in the revised version of the paper.**
>
> > Is the model less likely to provide incorrect information in high-stakes contexts, like asking for medical advice, than in a low-stakes context like asking which of two jokes is funnier?
>
> In response to your suggestion about examining high-stakes contexts (Q1), **we have added experiments with MedMCQA in two versions, with two and three options to choose**. These additions provide broader insights into how models handle the balance between instruction-following and accuracy across different contexts and stakes levels.
>
> > Is the model more likely to answer when there is an answer that is nearly correct versus all completely wrong?
>
> Regarding your concern about "close enough" answers, **we conducted a detailed analysis of answer patterns in the BAD dataset**. Our findings actually strengthen our original argument. We discovered that models choose the closer incorrect answer only in approximately 50% of cases. This random-like behavior suggests the problem is more serious than initially thought. The lack of consistent preference for closer answers indicates that models aren't making reasoned trade-offs between accuracy and instruction-following. We acknowledge that different use cases require different balances - for instance, API contexts might require strict instruction-following, while medical diagnosis should prioritize accuracy.
>
> > Users can instruct the language model to respond with a "non-of-the-above" option. If they do not, they may prefer instruction following over accuracy.
>
> Thank you for suggesting explicit 'none-of-the-above' options. Our research, building on studies [1] and [2], shows that adding this option actually reduces model performance. Rather than focusing on prompt engineering solutions, we believe addressing the core challenge of helping models recognize and reject flawed options would be more effective. We're interested in exploring alternative approaches that could improve model capability in this area.
>
> We appreciate this constructive feedback and are committed to ensuring our findings are presented as clearly and comprehensively as possible. Please don't hesitate to let us know if any aspects would benefit from further development or explanation.
>
> Citations:
>
> [1] Wang, H., Zhao, S., Qiang, Z., Xi, N., Qin, B., & Liu, T. (2024). Beyond the Answers: Reviewing the Rationality of Multiple Choice Question Answering for the Evaluation of Large Language Models. arXiv:2402.01349 (https://arxiv.org/abs/2402.01349)
>
> [2] Kadavath, S., Conerly, T., Askell, A., Henighan, T., Drain, D., Perez, E., Schiefer, N., Hatfield-Dodds, Z., DasSarma, N., Tran-Johnson, E., Johnston, S., El-Showk, S., Jones, A., Elhage, N., Hume, T., Chen, A., Bai, Y., Bowman, S., Fort, S., Ganguli, D., Hernandez, D., Jacobson, J., Kernion, J., Kravec, S., Lovitt, L., Ndousse, K., Olsson, C., Ringer, S., Amodei, D., Brown, T., Clark, J., Joseph, N., Mann, B., McCandlish, S., Olah, C., & Kaplan, J. (2022). Language Models (Mostly) Know What They Know. arXiv:2207.05221 (https://arxiv.org/abs/2207.05221)

---

> > ### Comment · Reviewer_Vdx7 · 2024-11-22
> >
> > > We believe the ideal approach is for the model to inform the user when there may not be a correct answer.
> >
> > I do not feel like your paper has a very nuanced view on this important topic. Many if not most chats are typed with mistakes--look at the wildchat or lmsys datasets. One of the best features of LLMs is that they can understand malformed and poorly worded questions. You ask for
> > ```
> > restraunt suggestions in newyork that serve lamb
> > ```
> > and you want a list of restaurants and not
> > ```
> > Here are the issues with the wording of the question:
> > 	1.	Misspelling:
> > 	•	“restraunt” should be “restaurant.”
> > 	•	“newyork” should be “New York.”
> > 	2.	Lack of Clarity:
> > 	•	The question does not specify whether it refers to New York City or the entire state of New York, which could lead to confusion.
> > 	3.	Inconsistent Capitalization:
> > 	•	Proper nouns like “New York” should always be capitalized.
> > 	4.	Lack of Specificity:
> > 	•	It does not mention whether it is looking for fine dining, casual eateries, or specific types of cuisine.
> > 	5.	Grammar:
> > 	•	The phrasing “restaurant suggestions in New York that serve lamb” is slightly awkward. A better construction might be, “Can you suggest restaurants in New York that serve lamb?”
> > 	6.	Context Missing:
> > 	•	The question does not indicate dietary preferences, budget, or location preferences within New York, which would provide useful context.
> > ```
> >
> > > For example, a model could respond with something like: "Wait, there are no good options provided, but if I must choose, I will select the closest one."
> >
> > I am trying to understand the motivation for multiple choice questions as a basis for this study. It seems to me that multiple choice questions are basically only used for evaluating models programmatically, that end users rarely type in multiple choice questions in normal use. Similar to a forced-choice scantron test that humans take where they need to fill in the bubbles, it is generally expected to give the "best guess" than write in an answer. I don't see your experiment as a great test of what chatbots should do when there is no right answer, which I imagine happens in real usage but in very different scenarios.

---

> > > ### Comment · Reviewer_Vdx7 · 2024-11-22
> > >
> > > > Thank you for suggesting explicit 'none-of-the-above' options. Our research, building on studies [1] and [2], shows that adding this option actually reduces model performance.
> > >
> > > If by performance, you mean accuracy, then yes it should reduce accuracy because the model will guess less. How ever it should also reduce error rates.

---

> ### Author Response · Authors · 2024-11-23
>
> Thank you for your thoughtful feedback. We apologize for not being sufficiently clear about our research scope and objectives in the manuscript. Let us clarify our focus:
>
> Our research examines scenarios where strict instruction-following must be balanced with intelligent oversight. For example, in tax document processing, forms have rigid structures and explicit instructions like "select only one box" or "if Line 1 exceeds X, complete Schedule A". When humans make errors by blindly following these instructions, LLMs should be able to flag potential inconsistencies rather than simply propagating the error. This is particularly critical in tax audits where the goal is ensuring compliance, not merely adhering to the form instructions.
>
> The central objective of our paper explores what happens when models prioritize strict instruction-following over reasoning or reflection, particularly in constrained formats. We observe that this has tangible consequences in domains requiring strict compliance with protocols.
>
> Our research contributes specifically to understanding:
> - models' behavior under rigid constraints, particularly their ability to recognize flawed scenarios while following strict instructions
> - how reflective judgment can serve as a safeguard against harmful consequences of blind adherence in constrained environments
>
> Regarding your concern about malformed inputs and subjective interpretation: we acknowledge these are critical challenges in the AI deployment. Our paper does not aim to solve these broad issues. Instead, we focus specifically on one aspect: how to maintain appropriate skepticism even within rigid instruction frameworks. This narrower focus allows us to contribute meaningful insights to the structured decision-making scenarios.
>
> We will revise our manuscript to better explain the motivation behind our research focus and its practical significance. Thank you for your thoughtful analysis - we appreciate your feedback and would be happy to address any additional questions you may have.

---

> > ### Author Response · Authors · 2024-11-25
> >
> > Dear Reviewer,
> >
> > As the ICLR rebuttal period will be ending soon, we are writing to follow up on our revised submission. We greatly appreciate your thorough feedback, which has guided us in enhancing our work through additional experiments and clarifications.
> >
> > We would be grateful to know if our revisions have adequately addressed your concerns. **If there are any remaining issues, we welcome your suggestions for improvements** that could strengthen our contribution and potentially lead to a higher evaluation score. We believe our work would make a valuable contribution to the ICLR community.

---

> > > ### Comment · Reviewer_Vdx7 · 2024-11-27
> > >
> > > Like other reviewers, I do not feel that multiple-choice questions gets at the heart of the interesting topic you raise: namely the tradeoff between following instructions and accuracy.

---

### Official Review · Reviewer_pEkD · 2024-11-02

**Soundness:** 3
**Presentation:** 2
**Contribution:** 2
**Rating:** 5
**Confidence:** 3

**Summary:**

The paper proposes the issue that large language models (LLMs) prioritizing instruction-following over critical reasoning. In this paper, They evaluate different LLMs through two datasets: Basic Addition Dataset (BAD) and MMLU Subset, where the given multiple-choice options are incorrect. Different LLMs were tested under different conditions (easy, standard, hard) that varied in instruction complexity. The authors measured Reflective Judgment Score (RJscore), defined as the proportion of instances where models demonstrated reflective judgment by refusing or correcting erroneous options.

**Strengths:**

1. the idea is interesting
2. they propose a new metric called Reflective Judgment metric  and also conduct a human study

**Weaknesses:**

1. I didn't see the difference between Reflective Judgment and refusal mechanism, in some sense I think they are the same. The authors should emphasize the differences between these two.

2.the data used to test different LLMs are quite limited. there are two weak points here: 1. The task (maybe I missed somewhere in the paper) lacks a version where partial given multiple-choice options are incorrect and partial given multiple-choice options are correct. 2. The  subset of MMLU and synthetic data (BAD), which may not fully generalize to diverse scenarios

3. overemphasize on the model size. Although larger models show better reflective judgment, this focus on scaling may overlook other aspects which may lead to these results.

**Questions:**

can you please explain how you obtain the confidence region in figure 2?

---

> ### Author Response · Authors · 2024-11-22
>
> Dear Reviewer,
>
> We greatly appreciate your thoughtful review of our work. We have carefully addressed each of your points below:
>
> > I didn't see the difference between Reflective Judgment and refusal mechanism, in some sense I think they are the same. The authors should emphasize the differences between these two.
>
> We acknowledge that this distinction could have been better articulated in our paper. Reflective judgment differs from traditional refusal mechanisms in several key ways:
>
> * Refusal mechanisms typically focus on identifying questions outside a model's knowledge domain or potentially harmful content.
> * Reflective judgment involves critically evaluating the validity of options even within the model's knowledge domain.
> * While refusal mechanisms are binary (answer/don't answer), reflective judgment includes the ability to provide correct answers even when they're not among the options.
>
> **We clarified this distinction in the revised version.**
>
> > the data used to test different LLMs are quite limited. there are two weak points here: 1. The task (maybe I missed somewhere in the paper) lacks a version where partial given multiple-choice options are incorrect and partial given multiple-choice options are correct. 2. The subset of MMLU and synthetic data (BAD), which may not fully generalize to diverse scenarios
>
> Thank you for highlighting these important concerns. For each dataset, we've included control experiments with both correct and incorrect options, marked as baseline, to ensure models can handle the questions and to measure any performance degradation. Our results show that models performing well on baseline questions often struggle when presented with incorrect options.
>
> Regarding the MMLU dataset, while we structured our subset to include 100 questions from each major domain (STEM, humanities, social sciences, business, medicine), we agree with your assessment about potential limitations in generalizability. In response to this feedback, **we have conducted additional experiments with MedMCQA**, which introduce **high-stakes medical scenarios**. These additions help demonstrate that reflective judgment challenges persist across different domains and complexity levels.
>
> > overemphasize on the model size. Although larger models show better reflective judgment, this focus on scaling may overlook other aspects which may lead to these results.
>
> Thank you for raising this concern about overemphasis on model size. While our initial results showed correlation between model size and reflective judgment ability, we agree that other factors deserve attention.
>
> Our new analysis reveals interesting differences: while both models have similar performance on baseline tasks (>83% accuracy), they show notably different patterns in reflective judgment. At the 7B scale, Qwen demonstrates reflective judgment abilities (around 20% RJ score) while Llama-8B consistently shows 0% RJ score. While these differences could be attributed to various factors including architecture, training data, or training methodology (which we cannot isolate due to limited access to training details), these results demonstrate that model size alone does not determine reflective judgment capabilities.
>
> > you please explain how you obtain the confidence region in figure 2?
>
> We thank you for pointing out the need for clarity about the confidence region calculations. **We have now added a complete explanation of how the confidence bands were computed to the main body of our paper**. The blue shaded region shows the 95% confidence band calculated using the standard formula: confidence_interval = t * SE * sqrt(1/n + (x - mean_x)^2 / sum_sq_x), where t is the t-value, SE is standard error, and n is sample size.
>
> We value your expertise and would be grateful for any additional guidance on how to better present these results. If you feel certain aspects need deeper examination, we're eager to address them.

---

> > ### Author Response · Authors · 2024-11-25
> >
> > Dear Reviewer,
> >
> > Understanding that you have many commitments, we are reaching out as the rebuttal period will end soon. We are truly grateful for your detailed review and have made substantial improvements based on your valuable feedback, including additional clarifications, revisions, and new experiments. **Please let us know whether we have addressed your concerns.** If we have not, we invite further suggestions that would allow you to increase your score. We believe our submission offers valuable contributions to the ICLR community.

---

> > > ### Comment · Reviewer_pEkD · 2024-11-26
> > >
> > > many thanks for the authors' clarification and for addressing all my concerns.

---

> > > > ### Author Response · Authors · 2024-11-27
> > > >
> > > > Dear Reviewer,
> > > >
> > > > Thank you for your positive feedback. **We are glad that our clarifications have successfully addressed all your concerns.** Given this, we would greatly appreciate if you could consider revising your score to reflect these improvements.
> > > >
> > > > Thank you for your time and thorough review of our work.
> > > >
> > > > Warmly,
> > > > The Authors

---

### Official Review · Reviewer_xowJ · 2024-11-03

**Soundness:** 2
**Presentation:** 3
**Contribution:** 2
**Rating:** 3
**Confidence:** 4

**Summary:**

The paper introduces the concept of "reflective judgment" in LLMs, which measures the ability to disobey instructions and reject answering or providing correct answers when given multiple-choice questions with only incorrect options.

The authors construct a simple dataset (Basic Addition Dataset (BAD), and present evidence that larger models and base models (before instruction tuning or alignment) exhibit better "reflective judgment". They also include limited experiments on the MATH dataset.

They also include a human study, where they find that the majority of human raters chose to follow instructions over applying "reflective judgment".

**Strengths:**

* Presentation: The paper is generally well written and easy to follow.
* Novelty: The authors study whether models can abstain in a multiple choice setting, where abstention creates a conflict because it would violate the given instructions.
* Broad evaluation:  The authors evaluate a large number of both open and closed LLMs, across different model sizes and training stages.
* Human study: The inclusion of a human study provides interesting insights into human responses in the same setting.

**Weaknesses:**

* Framing: I am not convinced the framing of the research question is useful. Whether a model limits its response to a set of options, or whether it rejects all available options, should depend on context and prompt (both underexplored, see below).
* Main dataset is very simple: The Basic Addition Dataset (BAD), consisting only of simple addition problems, is too narrow to draw strong conclusions about the general reflective judgment abilities of LLMs.
* Unclear impact of instruction tuning:  While the paper claims that instruction tuning hinders reflective judgment, the results are mixed. DeepSeekMath-7B exhibits decreased performance after instruction tuning, but Qwen2.5 does not. The paper doesn't sufficiently disentangle the effects of instruction tuning from alignment.
* Weak analysis of prompt variations: The analysis of prompt variations is superficial and lacks a deeper exploration of why certain prompts elicit more reflective responses.
* Limited exploration of CoT impact: While the paper notes the positive impact of CoT on reflective judgment in some cases, it lacks an investigation of the limitations and observed drawbacks of CoT in o1.

**Questions:**

1. Can you eloborate on the general framing of the research question of your paper. Specifically, you seem to suggest that "reflective judgment" is generally desirable. If you believe that's the case, can you more clearly justify it?
2. Can you further explore the relationship between instruction tuning, specific kinds of alignment and specific prompt formats and disentangle their individual effects on "reflective judgment" by also including a qualitative analysis? In other words, analyse whether certain prompts elicit an appropriate "reflective" responses if the model was trained to respond that way?
3. Can you provide a more detailed discussion about limitations and observed drawbacks of CoT in "reflective judgment".

---

> ### Author Response · Authors · 2024-11-22
>
> Dear Reviewer,
>
> Thank you for your thorough review and constructive feedback. We appreciate the time you've taken to evaluate our work and would like to address your concerns:
>
> > Main dataset is very simple: The Basic Addition Dataset (BAD), consisting only of simple addition problems, is too narrow to draw strong conclusions about the general reflective judgment abilities of LLMs.
>
> Thank you for raising the concern about dataset limitations. While BAD consists of simple arithmetic problems, we deliberately chose this as a starting point to isolate the reflective judgment behavior in a controlled, unambiguous setting. However, **we expanded our evaluation** significantly:
>
> * **We tested on MedMCQA**, involving high-stakes medical decisions, where we observed similar reflective judgment patterns
> * **We included more complex multiple-choice scenarios** (A, B, and C options) to verify these behaviors beyond binary choices
> * **Our analysis shows** that models choose closer incorrect answers only 50% of the time, suggesting fundamental decision-making patterns rather than simple numerical proximity bias
>
> > Can you eloborate on the general framing of the research question of your paper. Specifically, you seem to suggest that "reflective judgment" is generally desirable. If you believe that's the case, can you more clearly justify it?
>
> Thank you for raising the question about context-dependency and reflective judgment. We designed this study to examine how models handle situations with clearly incorrect options, particularly in critical scenarios. In medical diagnosis, for instance, choosing between two wrong diagnoses could be more harmful than acknowledging neither is correct. **Our results from MedMCQA testing support this concern**.
>
> We agree that not every situation requires rejection of options, but understanding when and how models question incorrect choices helps us build more reliable systems. This is especially relevant in high-stakes decisions where blind instruction-following could have serious consequences.
>
> > DeepSeekMath-7B exhibits decreased performance after instruction tuning, but Qwen2.5 does not. The paper doesn't sufficiently disentangle the effects of instruction tuning from alignment.
>
> Thank you for pointing out this important distinction. **We should clarify a key finding** in our results:
>
> **DeepSeekMath-7B-RLHF maintains high reflective judgment (100%)** while **Qwen2-Math-7B shows significant degradation after RLHF** (from 100% to 53%). **This difference appears linked to DeepSeekMath's unique training approach** - it consistently uses Chain-of-Thought reasoning regardless of user instructions.
>
> **This highlights two key trade-offs**:
> * CoT mostly help maintain reflective judgment through alignment training
> * However, DeepSeekMath's "always-on" CoT approach, while effective for reflective judgment, may not be ideal when users need concise answers
>
> **This suggests future work** might explore ways to preserve reflective judgment without requiring verbose CoT responses in all cases.
>
> > Can you provide a more detailed discussion about limitations and observed drawbacks of CoT in "reflective judgment".
>
> Thank you for this insightful question. **Our analysis compared DeepSeekMath-7B** in both instruction-tuned and RLHF versions, revealing distinct patterns:
>
> **The instruction-tuned version consistently uses a very rigid pattern**, starting each answer with "To solve the problem, we simply add..." followed by the calculation. While it always calculates the correct answer, it then forces an option selection without questioning, showing minimal reasoning depth despite the CoT prompt. **Most notably**, even when its calculation clearly differs from both options, it never questions this discrepancy.
>
> In contrast, **DeepSeekMath-7B-RLHF shows more variable behavior** with CoT prompting. In some cases, it identifies and explicitly questions incorrect options, saying things like "neither of the given options matches our result..." when calculating 8 + 9 = 17. However, this reflective behavior isn't consistent - in other problems it still forces choices despite calculating different answers.
>
> **Interestingly, GPT-4o-mini demonstrates** how CoT can effectively override the "solely A or B" instruction constraint. It first completes its reasoning, then compares the result with the options, and often refuses to select either when they don't match - showing how CoT can override strict instruction-following in some models.
>
> **This comparison suggests** that while RLHF training introduces some capacity for reflective judgment through CoT, this capability remains inconsistent. The instruction-tuned version, despite showing correct mathematical reasoning, seems to prioritize strict instruction-following over questioning incorrect options.
>
> We appreciate your thoughtful feedback and would welcome any additional perspectives on how to strengthen these findings in the paper.

---

> > ### Comment · Reviewer_xowJ · 2024-11-24
> >
> > Thank you for the detailed response!
> >
> > I remain unconvinced that the framing of the question is useful. In situations where the correctness of the answer is critical (like high-stakes medical decisions) asking the model to "Respond solely with A or B, even when neither option is correct" is highly problematic. You are basically measuring the effectiveness of instruction-following fine-tuning in an unrealistic setting, where the user tries to suppress disagreement from a model.
> >
> > For example, I think it would be more interesting to see if the model is biased by offering answer options.

---

> ### Author Response · Authors · 2024-11-25
>
> We agree on limitations of binary choices, especially in high-stakes scenarios. Let us clarify our approach: this research represents an initial step, drawing from current industry practices where conversational agents often operate within defined guidelines and specific response frameworks.
> While we acknowledge that forcing binary choices isn't ideal for all situations, our study aims to understand model behavior within common real-world constraints. Many applications (particularly commercial chatbots) require structured responses following specific guidelines. Understanding how models perform under these constraints provides valuable insights into their instruction-following capabilities.

---

### Official Review · Reviewer_1xP6 · 2024-11-04

**Soundness:** 2
**Presentation:** 3
**Contribution:** 2
**Rating:** 5
**Confidence:** 5

**Summary:**

This paper introduces the concept of "reflective judgment" in LLMs, defined as their ability to recognize and reject incorrect multiple-choice options rather than blindly following instructions. The authors evaluate various LLMs using both a BAD and a subset of MMLU, comparing their performance across different model sizes and training approaches. The study includes a parallel human study and analysis of how alignment techniques affect reflective judgment capabilities.

**Strengths:**

The paper identifies an important aspect of LLM behavior that warrants investigation - the tension between instruction-following and critical evaluation.

The experimental setup is systematic, testing multiple hypotheses: 1) Impact of model size on reflective judgment. 2) Effect of alignment techniques. 3) Relationship with chain-of-thought reasoning

The inclusion of a human study provides valuable comparative insights into how humans handle similar decision-making scenarios.

**Weaknesses:**

The dataset and evaluation design present several fundamental limitations. The BAD dataset's restriction to basic addition operations significantly constrains the generalizability of the findings. The binary choice format (A/B) represents an oversimplified decision-making scenario that may not reflect real-world complexity. Furthermore, the MMLU subset of 400 questions may not be sufficiently representative of broad domain knowledge, and the evaluation lacks complex reasoning scenarios where reflective judgment would be most crucial.

While the human study is appreciated, I feel these type of experiments could be biased. The human study's small sample size of 50 participants restricts the statistical significance of the findings. The experimental design does not control for question order effects, and the exploration of prompt variations is limited. The absence of ablation studies makes it difficult to isolate which factors contribute most significantly to reflective judgment capabilities.

The concept of "reflective judgment" needs stronger grounding in existing literature, and the paper provides insufficient exploration of how different contexts might affect this capability. The analysis of why alignment techniques impact reflective judgment remains superficial, leaving important questions about the relationship between alignment and critical thinking unanswered.

While the findings show significant improvements (>85%) with chain-of-thought reasoning, the paper does not adequately investigate the underlying mechanisms. The interaction between model architecture and reflective judgment capabilities remains unexplored, and the potential trade-offs between helpfulness and critical thinking need deeper examination. The relationship between model size and reflective judgment, while observed, lacks theoretical explanation.

The paper notably lacks practical solutions or mitigation strategies. While it identifies important challenges in maintaining reflective judgment during alignment, it does not propose concrete approaches to address these issues. The absence of practical recommendations for improving model training and evaluation methods limits the immediate applicability of the findings. Without suggested solutions, the paper's contribution to advancing the field remains primarily diagnostic rather than prescriptive.

**Questions:**

Could you provide grounding for the "reflective judgment" concept in existing literature?

What mechanisms might explain the relationship between model size and reflective judgment ability?

How might the findings generalize to more complex reasoning tasks beyond arithmetic?

Could you elaborate on why alignment techniques appear to impair reflective judgment?

What design considerations would be important for a more comprehensive evaluation dataset?

How do different chain-of-thought prompting strategies affect reflective judgment?

Could you propose potential solutions for maintaining reflective judgment during alignment?

---

> ### Author Response · Authors · 2024-11-22
>
> Dear Reviewer,
>
> Thank you for your thorough review and thoughtful questions. We appreciate your recognition of the importance of studying the tension between instruction-following and critical evaluation in LLMs. **We have substantially revised our paper** to address your concerns about methodology, theoretical grounding, and practical applications.
>
> > Could you provide grounding for the "reflective judgment" concept in existing literature?
>
> We appreciate your insightful question regarding the grounding of the "reflective judgment" concept in existing literature. Our work was inspired by the Reflective Judgment Model as developed by King and Kitchener (1994) [1], which itself builds upon foundational observations by Dewey (1933) [2] regarding reflective thinking. This model describes the development of reasoning and critical thinking, particularly in contexts where problems lack clear-cut solutions or involve epistemic uncertainty.
>
> In our work, we extend this concept to the domain of large language models (LLMs). Specifically, we introduce reflective judgment as a metric to evaluate an LLM's ability to critically assess and respond to flawed or incomplete instructions. Unlike traditional refusal mechanisms that operate on predefined boundaries, reflective judgment in LLMs examines their capacity to recognize and articulate the invalidity of presented options.
>
> > What design considerations would be important for a more comprehensive evaluation dataset? How might the findings generalize to more complex reasoning tasks beyond arithmetic?
>
> Our evaluation framework prioritizes diverse task types, ranging from basic instruction-following to complex reasoning and judgment tasks. **We included** ambiguous/invalid scenarios to test rejection capabilities, and real-world applications through datasets like MedMCQA covering medical specialties. Our findings confirm that models struggle to question incorrect options even when making important decisions like medical diagnosis, showing this is a deep problem, not just limited to simple tasks.
>
> > Could you elaborate on why alignment techniques appear to impair reflective judgment?
>
> Alignment techniques like RLHF and supervised fine-tuning aim to make LLMs more helpful, safe, and aligned with user intentions. However, our findings reveal trade-offs that impact critical reasoning. Highly aligned models often prioritize instruction-following, even when provided options are incorrect. This reflects an overemphasis on compliance, limiting their ability to reject flawed inputs.
>
> Bias in training data, typically featuring well-structured queries with clear answers, further reinforces this tendency. Our human evaluation revealed similar challenges with invalid options, suggesting RLHF-based alignment may propagate human cognitive biases into models.
>
> > While the human study is appreciated, I feel these type of experiments could be biased. The human study's small sample size of 50 participants restricts the statistical significance of the findings.
>
> We appreciate your thorough feedback on our human study and agree that additional methodological details would strengthen our presentation. **We have added** our complete experimental protocols to Appendix E, detailing the randomized question order and distribution. Our power analysis indicates that 50 participants provided sufficient statistical power (85%) to detect medium-sized effects (10-15% differences) at p < 0.05.
>
> > How do different chain-of-thought prompting strategies affect reflective judgment?
>
> **We have clarified** our analysis of why chain-of-thought (CoT) prompting improves reflective judgment. For STEM questions, particularly in mathematics, CoT prompting encourages models to first calculate the correct answer independently before comparing it to the provided options. When models find no matching option, they are more likely to flag this discrepancy.
>
> However, this improvement is domain-specific. In areas such as law or medicine, where answers require complex domain knowledge rather than calculation, CoT provides minimal benefit. This limitation highlights that while CoT can enhance performance in structured problems, it is not a universal solution.
>
> > Could you propose potential solutions for maintaining reflective judgment during alignment?
>
> **We have added in Limitation and Future Work** a substantial section on practical solutions:
>
> * Modified reward modeling that explicitly values appropriate refusal
> * Training protocols that balance instruction-following with accuracy
> * Training the model to always respond in CoT manner
> * Our dataset analysis reveals that better datasets may be needed
>
> We appreciate your thoughtful feedback and would be happy to further discuss or clarify any aspects of these findings.
>
> Citations:
>
> [1] King, P., Kitchener, K. & Wood, P. (1994). Developing reflective judgment. San Francisco: Jossey-Bass.
>
> [2] Dewey, J. (1933). How We Think. Boston: D.C. Heath & Co Publishers.

---

> > ### Comment · Reviewer_1xP6 · 2024-11-22
> >
> > Thank you for your detailed response. Your clarifications on theoretical grounding, human study methodology, and proposed solutions have improved the paper, and I am increasing my score to 4. However, key limitations remain, especially the dataset limitations (binary choice format, basic arithmetic focus) still restrict the generalizability of findings. Besides, the exploration of reflective judgment in complex reasoning scenarios needs further development. Given these core limitations, I suggest significant revisions would be needed to establish robust conclusions about reflective judgment in LLMs.

---

> > > ### Author Response · Authors · 2024-11-22
> > >
> > > > However, key limitations remain, especially the dataset limitations (binary choice format, basic arithmetic focus) still restrict the generalizability of findings. Besides, the exploration of reflective judgment in complex reasoning scenarios needs further development. Given these core limitations, I suggest significant revisions would be needed to establish robust conclusions about reflective judgment in LLMs.
> > >
> > > Thank you for your thoughtful feedback. We would like to clarify that our current work already addresses several key concerns, as **detailed in Section 4**:
> > >
> > > 1. **We extended our analysis** beyond simple arithmetic and STEM to **high-stakes domains through the MedMCQA dataset** (200 questions across Anesthesia, Pathology, Radiology, Surgery)**. While we used binary choices to establish baseline measurements, **we also tested increased complexity** by expanding to three options in medical scenarios.
> > >
> > > 2. **Our findings reveal that even in high-stakes medical decisions, LLMs struggle with reflective judgment, regardless of the number of options provided.** We were surprised to discover that even in such critical decision-making contexts, LLMs blindly follow instructions without proper critical evaluation.
> > >
> > > 3. **We investigated the nature of choices made by LLMs**. Our analysis shows that models choose closer incorrect answers only 50% of the time, suggesting fundamental decision-making patterns rather than simple numerical proximity bias.
> > >
> > > 4. **Our new probability analysis** (Figure 6) reveals how warnings affect model confidence differently across architectures - GPT-4o becomes more confident with warnings (showing negative gap in log probability up to -0.4) while GPT-4o-mini shows decreased confidence (positive gap up to 0.3). This indicates that simple warning systems might not work as intended, especially for larger models.
> > >
> > > We agree that future investigation into more complex scenarios would be valuable.
> > >
> > > We appreciate your comments and are happy to further improve the text or answer any questions.

---

> ### Author Response · Authors · 2024-11-25
>
> Dear Reviewer,
>
> Acknowledging your time constraints, we wanted to reach out regarding our revised submission as the rebuttal period will end soon. We are truly thankful for your comprehensive review and your decision to increase the score. However, we noticed you mentioned increasing it to 4, while the ICLR scoring system uses the scale of 1, 3, 5, 6, 8, and 10. Could you please clarify which score on this scale you intended to give?
>
> We greatly appreciate your positive feedback about our clarifications on theoretical grounding, human study methodology, and proposed solutions. **If there are any remaining concerns, we welcome further suggestions that would help improve the paper.** We are committed to ensuring our work makes a meaningful contribution to the ICLR community.

---

> > ### Author Response · Authors · 2024-11-27
> >
> > Dear Reviewer,
> >
> > We are following up on our previous communication regarding the rebuttal for our ICLR submission. We appreciate your decision to increase our score and would greatly value your response to our questions and any additional feedback you may have.
> >
> > Your insights are valuable to us, and we look forward to your response.
> >
> > Thank you for your time and consideration.
> >
> > Warmly,
> > The Authors

---

> > > ### Comment · Reviewer_1xP6 · 2024-12-02
> > >
> > > In response to your detailed response, I have updated my score. I suggest the revision to account for the rebuttals.

---

### Author Response · Authors · 2024-11-22

Dear Reviewers and Area Chair,

Thank you for taking the time to thoroughly review our paper and provide insightful feedback. We are encouraged that the reviewers recognized **the importance of our work on LLMs' instruction-following versus critical evaluation** (Reviewers 1xP6, Vdx7), praised our **systematic experimental approach** including model size analysis and human studies (Reviewers 1xP6, xowJ), and found the **presentation clear and well-structured** (Reviewer xowJ). The broad evaluation across multiple models and our **novel Reflective Judgment metric** were also highlighted as key strengths (Reviewers xowJ, pEkD).

**After careful consideration of your comments, we have made several significant revisions to strengthen the paper**:

1. **We expanded our analysis of LLM behavior across different complexity scenarios in Section 4 (Additional Analysis)**:
   * Added evaluation using MedMCQA dataset (200 questions spanning Anesthesia, Pathology, Radiology, and Surgery)
   * Investigated performance with increased task complexity by varying the number of options**
   * Analyzed language patterns when models encounter multiple incorrect choices

2. **We conducted new analyses on model uncertainty in Section 4:**
   * Investigated how different prompt structures affect model uncertainty

3. **To accommodate these additions while meeting the 10-page limit:**
   * We added examples of models' answers to Chain-of-Thought (CoT) prompting to Appendix F
   * Restructured the content to maintain clear flow and readability

We believe these changes comprehensively address the reviewers' feedback while strengthening the paper's contributions.

We appreciate your thoughtful guidance in improving this work and look forward to your feedback on these revisions.

Warmly,
The Authors

---

### Note · Authors · 2024-12-05

I have read and agree with the venue's withdrawal policy on behalf of myself and my co-authors.